# How social distancing, mobility, and preventive policies affect COVID-19 outcomes: Big data-driven evidence from the District of Columbia-Maryland-Virginia (DMV) megaregion

Jina Mahmoudi[1], Chenfeng Xiong[1,2]*

1 Maryland Transportation Institute, Department of Civil and Environmental Engineering, University of Maryland, College Park, Maryland, United States of America, 2 Shock Trauma Anesthesiology Research Center, School of Medicine, University of Maryland, Baltimore, Maryland, United States of America

* cxiong@umd.edu

**Data Availability Statement:** All data has been published open source via Github Repository: https://github.com/umdcxiong/social_distancing_DMV.

## Abstract

Many factors play a role in outcomes of an emerging highly contagious disease such as COVID-19. Identification and better understanding of these factors are critical in planning and implementation of effective response strategies during such public health crises. The objective of this study is to examine the impact of factors related to social distancing, human mobility, enforcement strategies, hospital capacity, and testing capacity on COVID-19 outcomes within counties located in District of Columbia as well as the states of Maryland and Virginia. Longitudinal data have been used in the analysis to model county-level COVID-19 infection and mortality rates. These data include big location-based service data, which were collected from anonymized mobile devices and characterize various social distancing and human mobility measures within the study area during the pandemic. The results provide empirical evidence that lower rates of COVID-19 infection and mortality are linked with increased levels of social distancing and reduced levels of travel—particularly by public transit modes. Other preventive strategies and polices also prove to be influential in COVID-19 outcomes. Most notably, lower COVID-19 infection and mortality rates are linked with stricter enforcement policies and more severe penalties for violating stay-at-home orders. Further, policies that allow gradual relaxation of social distancing measures and travel restrictions as well as those requiring usage of a face mask are related to lower rates of COVID-19 infections and deaths. Additionally, increased access to ventilators and Intensive Care Unit (ICU) beds, which represent hospital capacity, are linked with lower COVID-19 mortality rates. On the other hand, gaps in testing capacity are related to higher rates of COVID-19 infection. The results also provide empirical evidence for reports suggesting that certain minority groups such as African Americans and Hispanics are disproportionately affected by the COVID-19 pandemic.

**Funding:** The author(s) received no specific funding for this work.

**Competing interests:** The authors have declared that no competing interests exist.

# Introduction

## Background

The outbreak of a respiratory illness caused by a novel coronavirus (2019-nCoV) has plagued the world since December 2019. The outbreak first started in the city of Wuhan, China, but it rapidly spread internationally [1]. As a growing number of countries reported confirmed cases, the illness became known as the COVID-19 disease throughout the world. By March 11, 2020, the World Health Organization (WHO) announced the COVID-19 outbreak a pandemic [2] due to the disease reaching various countries in various continents. The United States is among the countries most affected by the COVID-19 pandemic. On January 21, 2020, the Centers for Disease Control and Prevention (CDC) confirmed the first case of the novel coronavirus (2019-nCoV) in the U.S. [3]. This first confirmed case was a resident of the state of Washington who had recently traveled to Wuhan, China. As of January 10, 2021, a total of 22,139,598 confirmed cases of COVID-19 and 372,552 related deaths have been reported in the U.S. [4].

Considering these statistics, identification of the factors that help slow the spread of the COVID-19, and thereby reduce the number of human deaths caused by it is undoubtedly of utmost importance. Among individual-level factors, older age and having an underlying medical condition are considered key risk factors for worse COVID-19 outcomes including death [5–8]. In addition to age, other sociodemographic attributes such as race and gender appear to also play a role in transmission and contraction of COVID-19 and its outcomes [5–10].

Further, as the 2019-nCoV virus is transmitted from person-to-person, measures to limit face-to-face contact with other individuals are among the main factors to reduce the transmission of COVID-19 [11, 12]. These preventive measures can be practiced at both the individual and community levels. Particularly, measures of physical/social distancing (e.g., staying six feet apart from other individuals, closure of non-essential businesses and facilities, closure of educational institutions, banning gatherings of large crowds), travel restrictions (e.g., stay-at-home orders, limitation of travel to essential travel only, teleworking), and other preventive measures (e.g., wearing face coverings by individuals, policies to mandate face coverings in public spaces) have a potential to reduce the transmission of COVID-19 [5, 11–14].

Each of the above measures alone or in combination with other measures can be effective in delaying the spread of COVID-19. Social distancing measures aim to slow the spread of COVID-19 by stopping chains of person-to-person transmission of the disease and preventing new ones from occurring [14]. Since the start of the COVID-19 outbreak in the U.S., many local governments and states have enacted policies to promote social distancing. These include declaration of state of emergency, issuance of stay-at-home/shelter-in-place orders, closure of non-essential businesses, closure of schools, and requirement of using face coverings/masks in public spaces. As a result, a substantial amount of social distancing has been practiced by Americans during the pandemic [11].

Travel restriction measures such as stay-at-home orders and promotion of teleworking can also help reduce the transmission of COVID-19 by reducing human mobility and limiting face-to-face interactions between individuals. The potential of teleworking to reduce the number of infections has been greatly employed during the COVID-19 pandemic as research has found that the share of the U.S. workforce working entirely from home increased from 8% in February 2020 to 35% in May 2020 [15].

Measures of social distancing and travel restrictions appear to be effective strategies in containing the spread of COVID-19 [16, 17]. However, premature relaxation of these measures can lead to an increase in the number of infections or even a second peak [16]. Therefore, the time of lifting of the social distancing and travel restriction measures—or in another words— the reopening time, can also be considered a key factor in containing the spread of COVID-19.

Among other preventive measures that can slow the spread of COVID-19 is using face coverings/masks in public settings. This practice can particularly be an important strategy to protect against contraction of COVID-19 where social distancing is difficult to maintain [5].

Further, enforcement is a crucial factor with respect to government-imposed measures, particularly those related to social distancing and travel restrictions. For instance, although at least 37 states, 74 counties, 14 cities, the District of Columbia (D.C.), and Puerto Rico had issued stay-at-home orders by April 1, 2020 [18], different approaches were taken by different jurisdictions to enforce these orders. As a result, there has been a wide variation in enforcement severity for noncompliance with stay-at-home orders by local governments and states. These range from educational measures to civil or even criminal penalties. One report suggested that by April 3, 2020, fifteen states (including D.C.) had stated that individuals found in violation of their stay-at-home orders would be subject to fines and/or confinement in jail; fourteen had stated that they would enforce the order by first issuing a warning, then possibly issuing a fine for a repeated offense; and fourteen more either had not stated how the order would be enforced or had explicitly stated that it would only be enforced through educational approaches with citizens, without penalties [19]. These variations in enforcement severity can influence individuals' compliance with of stay-at-home orders (or other government-imposed social distancing measures), and thereby affect their effectiveness in slowing the spread of COVID-19. For example, risking a fine of $1,000 or higher amounts and possible jail time for violating stay-at-home orders, which was implemented in several states including Maryland, New Jersey, Hawaii, as well as the District of Columbia, might have deterred residents of these states from noncompliance more so than residents of states with less severe enforcement strategies such as taking educational approaches with violators of the orders.

Other important factors that can play an important role in COVID-19 outcomes are measures of hospital capacity and testing capacity. Hospital systems are designed for average patient loads, not pandemics; therefore, if the system is stretched beyond its capacity, it can lead to dire situations such as having to make rationing decisions that can affect all patients including those with COVID-19 [20]. Crucial to better caring for hospitalized COVID-19 patients and preventing worse outcomes including death is access to scarce critical care resources such as ventilators and Intensive Care Unit (ICU) beds [20–22]. Testing capacity is also a key factor in delaying the spread of COVID-19. A higher testing capacity allows earlier identification of infected cases, which in turn allows quicker and more effective contact tracing, and can ultimately prevent new infections. Widespread testing has been suggested to be an effective strategy to prevent the resurgence of the disease [23, 24]. On the other hand, a lack in sufficient and efficient testing capabilities can lead to an increased risk of spread of the virus by infected individuals whose infection goes undetected due to gaps in testing capacity.

As the COVID-19 pandemic is rapidly evolving, research on the factors that affect the spread and outcomes of this new disease can be considered to still be in its infancy. Many gaps exist in research with respect to the role of various factors discussed in the preceding paragraphs in transmission of COVID-19 and the severity of its outcomes. Given these gaps in empirical research, this paper contributes to the existing body of knowledge on COVID-19 by:

a. examining the link between measures of social distancing, human mobility, enforcement strategies and strictness, and hospital and testing capacities in COVID-19 outcomes; and

b. using longitudinal data and modeling techniques to analyze the impact of such factors on COVID-19 outcomes, thereby providing preliminary evidence on the causality of the observed correlations.

The analysis benefits from usage of a large-scale, location-based service dataset that provides anonymized mobile device data on social distancing as well as mobility changes in a major megaregion during the COVID-19 pandemic. The big data-driven evidence presented in this study identifies a few factors that play an essential role in slowing the spread of COVID-19 spread and its worse outcomes. These findings can assist policy decision-makers to develop and implement more effective policies to reduce healthcare cost, overcome healthcare disparities, and prevent human loss of life during a public health crisis such as a global pandemic.

## Literature review

Many factors have been suggested by previous research to play a role in COVID-19–related outcomes. Among these are health status (e.g., presence or absence of an underlying medical condition), sociodemographic characteristics (e.g., age, gender, and race), as well as observation of preventive measures—at both the individual and community levels. At the individual level, the latter can include practicing physical distancing (i.e., staying six feet apart from other people) and/or wearing of a face covering, whereas at the community level, it can include having policies in place to promote social distancing and wearing a face mask in public spaces.

Existing literature suggests that underlying medical conditions such as hypertension, diabetes, chronic lung disease, cardiovascular disease, obesity, chronic kidney disease, asthma, and cancer can put individuals at a higher risk of severe illness from COVID-19 [5–8, 25].

Further, research thus far identifies older age as a key vulnerability factor for more severe COVID-19 outcomes [5–8]. The Centers for Disease Control and Prevention (CDC) also suggests that the risk of severe illness from COVID-19 increases with age and the greatest risk for severe illness from COVID-19 is among individuals aged $\geq$ 85 years [25]. In addition, males have been suggested by some studies to be affected by adverse COVID-19 outcomes at higher rates than females [5–8, 10]. For instance, Jin et al. [10] found that COVID-19 patients who were male were more at risk for worse outcomes including death, independent of age.

With respect to race/ethnicity, the existing literature suggests that minority groups have been disproportionally affected by the COVID-19 pandemic. The most pervasive disparities are observed among African American and Hispanic populations. These disparities are becoming more evident as more studies report higher rates of adverse COVID-19–related outcomes for these minority groups [5–7, 9]. For instance, in a study of the hospitalization rates for COVID-19 patients, Garg et al. [5] reported that while African Americans represented 18% of the study catchment population, they comprised 33% of the hospitalized COVID-19 patients, which indicated that the African American population may be disproportionately impacted by COVID-19 [5]. Other studies also suggest that African Americans are contracting COVID-19 at higher rates and are more likely to lose their lives from it [7]. An analysis conducted by Thebault et al. [9] in April 2020 indicated that counties with a majority population of African Americans had three times the rate of COVID-19 infections and approximately six times the rate of deaths compared to counties with a majority population of white residents. Further, the same study showed that in various U.S. jurisdictions, African Americans accounted for higher percentages of COVID-19 deaths than their representation percentages in the population. For example, the analysis indicated that approximately 70% of COVID-19 deaths in Chicago and 60% of COVID-19 deaths in District of Columbia involved African Americans, whereas they made up only about 30% and 45% of the population of these cities, respectively [9]. This is while Hooper et al. [6] suggested that by May 6, 2020, rates of COVID-19 confirmed cases in Chicago were greatest among Hispanics and African Americans, respectively, whereas the rate of COVID-19 deaths was

substantially higher among African Americans and Hispanics, respectively. The same study reported that by May 7, 2020, New York City had a greater age-adjusted COVID-19 death rate among Hispanics and African Americans, compared with white residents [6]. Other literature also suggested that in New York City, the disproportionate COVID-19 burden was validated for minority groups—especially, for African Americans and Hispanics—who with a population representation of 22% and 29%, respectively, accounted for 28% and 34% of COVID-19 deaths, respectively [7].

Observation of preventive measures and practices as well as adoption of preventive public health policies have also been suggested by existing literature as effective strategies to slow the spread of COVID-19. For instance, from their findings, Garg et al. [5] underscored the importance of preventive measures such as social distancing and wearing face coverings in public settings to protect vulnerable populations (e.g., older adults and those with underlying medical conditions) as well as the general public from COVID-19. Other research suggested that adherence to preventive practices such as use of face masks in public spaces, social distancing, and physical isolation can reduce transmission of COVID-19, flatten the curve of new cases, and save lives [7]. Empirical findings also showed that social distancing was an effective tool in reducing the number of COVID-19 infections in China [16]. Moreover, from a study of 11 European countries, Flaxman et al. [13] estimated that social distancing measures may have prevented as many as 59, 000 COVID-19 deaths through March 31, 2020. A study that evaluated the impact of social distancing measures on the growth rate of confirmed COVID-19 cases across the U.S. showed that adoption of government-imposed social distancing policies reduced the daily growth rate of confirmed COVID-19 cases [17].

One aspect of social distancing that emerges from the literature to play a role in reducing the spread of COVID-19 is its impact on travel. Literature suggests that policies enacted by local or state governments mandating social distancing as well as stay-at-home orders can lead to fewer trips by individuals. For example, Andersen [11] concluded that mandatory measures to increase social distancing—and most notably stay-at-home orders—were effective policies to decrease visits to locations outside of home. The reduction of travel due to mandatory social distancing policies and stay-at-home orders can reduce the risk of exposure to COVID-19 and aid in slowing the transmission of the disease. Previous research argued that combined social distancing policies and travel restrictions helped in lowering the spread of COVID-19 over the course of the ongoing outbreak in Wuhan, China [16].

Another potentially effective strategy to slow the transmission of COVID-19 is to reduce human mobility by imposing travel restriction policies. International and domestic travel restrictions have been shown to delay the spread of the disease [26, 27]. Other research concluded that the effect of stay-at-home orders negatively contributed to human mobility measured by the daily average number of trips and daily average person-miles traveled [28]. Other travel restrictions such as policies requiring teleworking (instead of travel to the place of work) have been suggested by past research to help reduce the number of COVID-19 infections by reducing face-to-face interactions between workers [15].

While reduction of human mobility can theoretically help in containment or mitigation of COVID-19, the impact of human mobility measures such as the number of trips, person-miles traveled, and teleworking on COVID-19 outcomes have not been thoroughly investigated in the past and further empirical research is warranted in this area. The present study aims to contribute to the empirical knowledge by using big mobile-device data providing information on human mobility changes during the COVID-19 pandemic within a megaregion that contains the U.S. capital city.

## Materials and methods

### Data

The study area for this analysis includes the District of Columbia in addition to various counties within the states of Maryland and Virginia. The study area is referred to as the DMV area (i.e., District of Columbia, Maryland, and Virginia) in the remainder of this paper. Data from several sources have been utilized to develop the database for the analysis. These include, but are not limited to, the University of Maryland COVID-19 Impact Analysis Platform [29] and the Johns Hopkins University COVID-19 Data Repository [30, 31]. These sources provide data on various characteristics of the DMV area including key human mobility and social distancing measures such as the average number of daily person trips and the percentage of residents staying at home [29] as well as key COVID-19 measures such as the numbers of confirmed COVID-19 cases and related deaths [30].

Daily mobile-device data on human mobility and social distancing were only available for the time period between January 1, 2020 and June 10, 2020. Therefore, this study is conducted using data within that specific period of time. After proper deduplication process, the average sampling rates for the three states are above 80%, indicating over 12 million devices were observed in our data pool [29, 30].

Figs 1 and 2 show the total number of confirmed COVID-19 cases and total related deaths per 100,000 population in the states within the DMV megaregion by the end of June, 2020. Due to the county being the smallest geographic area for which COVID-19 data are available, the unit of analysis in this study has been considered as the county. Figs 3 and 4 show the total number of confirmed COVID-19 cases and related deaths per 100,000 population for the counties within the DMV states by the end of June, 2020.

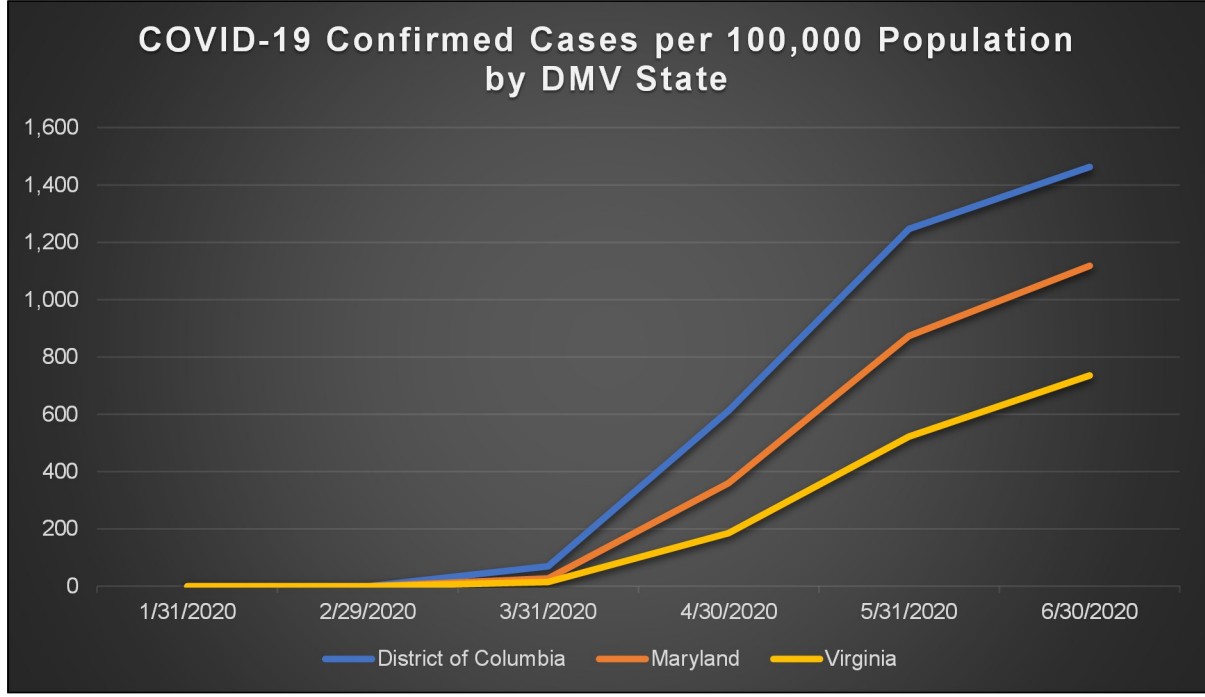

**Fig 1. COVID-19 confirmed cases per 100,000 population by DMV state.**

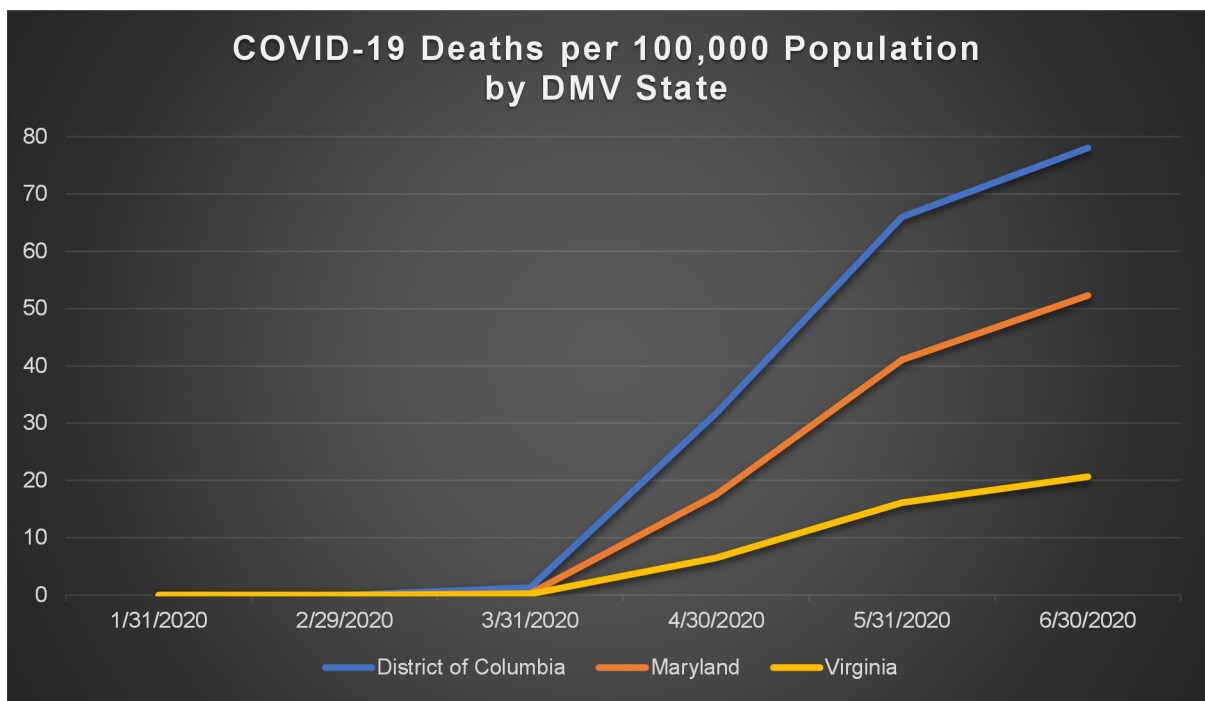

**Fig 2. COVID-19 deaths per 100,000 population by DMV state.**

Fig 3 indicates that by the end of June, 2020, the DMV counties with the highest number of confirmed COVID-19 cases per 100,000 population were Prince George's County in Maryland as well as Accomack County, Buckingham County, Greensville County, Northampton County, Richmond County, and Emporia City in Virginia. Fig 4 indicates that by the end of June, 2020, the DMV counties with the highest number of COVID-19 deaths per

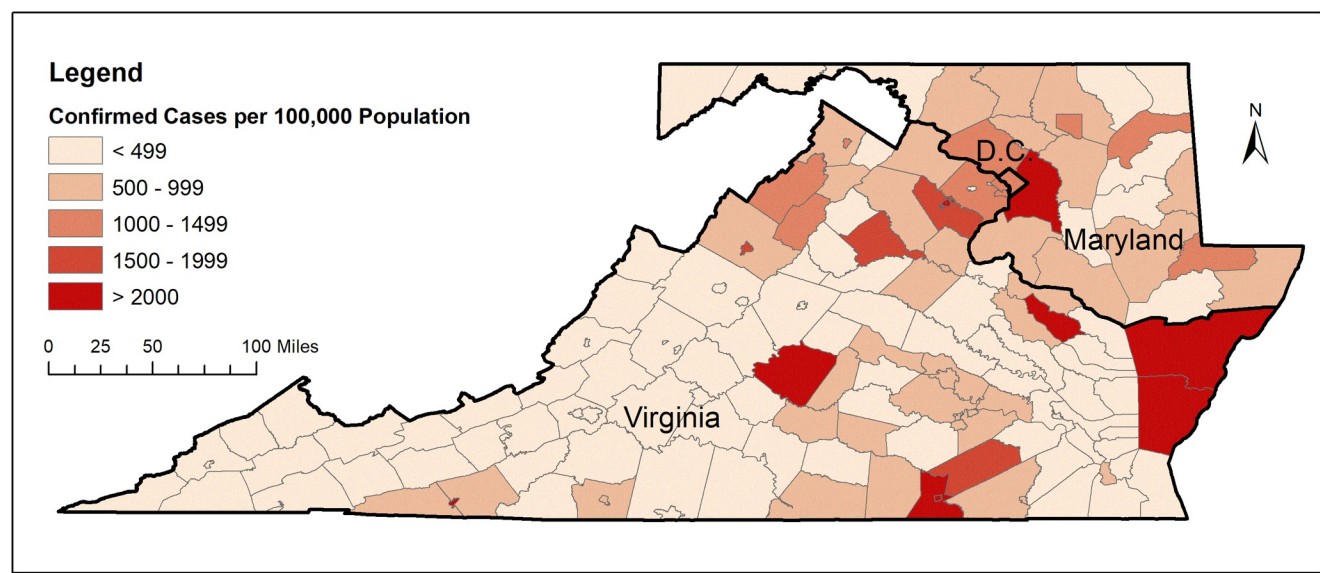

**Fig 3. COVID-19 confirmed cases per 100,000 population for counties within the DMV area.** Source of data: COVID-19 Data Repository by the Center for Systems Science and Engineering at Johns Hopkins University [30]. Data by June 30, 2020.

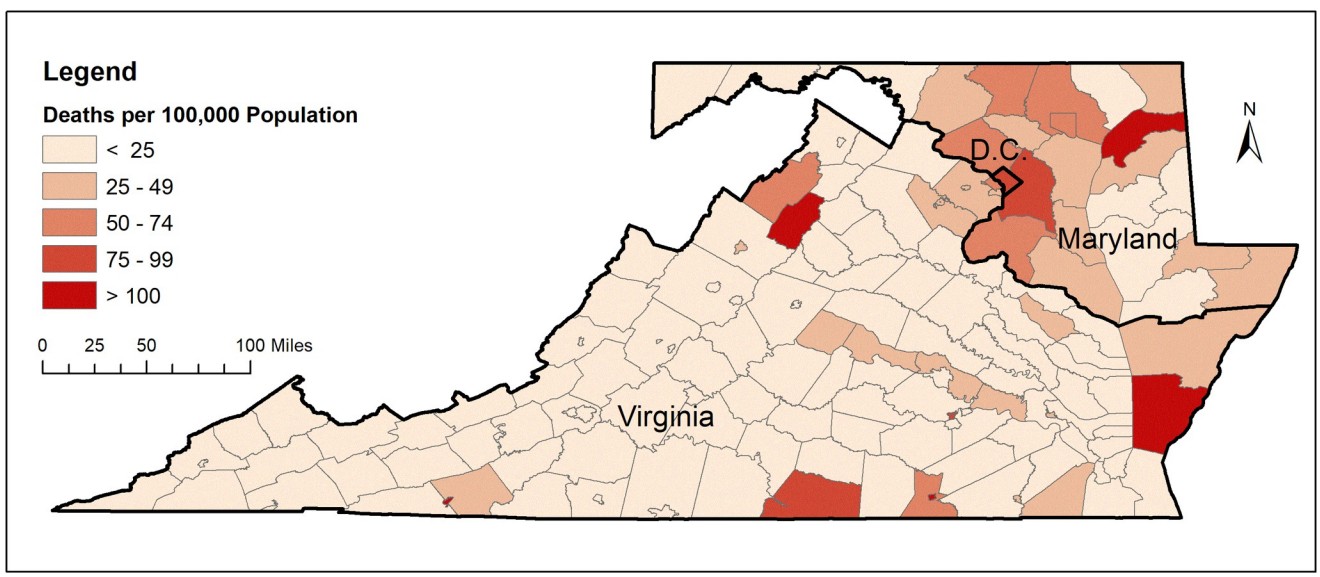

**Fig 4. COVID-19 deaths per 100,000 population for counties within the DMV area.** Source of data: COVID-19 Data Repository by the Center for Systems Science and Engineering at Johns Hopkins University [30]. Data by June 30, 2020.

100,000 population were Kent County and Prince George's County in Maryland as well as Mecklenburg County, Northampton County, and Page County in Virginia. District of Columbia was also among the DMV jurisdictions with a high number of COVID-19 deaths per 100,000 population, as seen from Fig 4.

## Modeling techniques

Due to the database used for the analysis being a panel (i.e., longitudinal) database, panel data modeling techniques have been employed in this study to model COVID-19–related outcomes. Panel data track particular units over a period of time. In another word, each unit is observed in the data set at various points in time. Modeling multiple observations on each unit can provide more accurate estimates and insights into causality of effects as compared to cross-sectional models, which can only capture correlations.

According to Wooldridge [32], a linear panel-data model can be formulated as Eq 1, where for each cross section unit, data are observed on the same set of variables for $T$ time periods:

$$y_{it} = \theta + \mathbf{X}_{it}\beta_1 + \mathbf{K}_i\beta_2 + c_i + \varepsilon_{it} \quad i = 1, 2, \ldots, N \text{ and } t = 1, 2, \ldots, T \qquad (1)$$

where,

$y_{it}$ is the dependent variable;

$\mathbf{X}_{it}$ is a $(1 \times P_1)$ vector of time-varying independent variables;

$\mathbf{K}_i$ is a $(1 \times P_2)$ vector of time-invariant independent variables;

$\theta, \beta_1$ and $\beta_2$ are model parameters;

$c_i$ is a time-invariant unobserved unit-specific effect; and

$\varepsilon_{it}$ is a time-varying idiosyncratic error term.

The assumptions for the models developed in this study are as follows:

a) strict exogeneity:  $\mathrm{E}\left(\varepsilon_{it} | \mathbf{X}_i, \mathbf{K}_i, c_i\right) = 0 \quad t = 1, 2, \ldots, T$ (2)

b) independence:  $\{\mathbf{X}_i, \mathbf{K}_i, y_i\}_{i=1}^N$

$$\text{c) serial correlation: } \mathrm{E}\left(\varepsilon_{it}|\varepsilon_{is}\right) \neq 0 \text{ for all } s \neq t \tag{3}$$

$$\text{d) panel-level heteroskedasticity: } \mathrm{Var}\left(\varepsilon_i|\mathbf{X}_i,\ \mathbf{K}_i\right) = \sigma_i^2 \mathrm{I}_\mathrm{T} \tag{4}$$

## Model variables

Two COVID-19 outcomes have been considered is this study for modeling purposes: the daily rate of confirmed COVID-19 cases (i.e., infections) within the county, and the daily rate of COVID-19 deaths within the county. These outcomes are quantified in the models by the following two dependent variables: the daily cumulative number of confirmed COVID-19 cases per 1,000 county population, and the daily cumulative number of deaths from COVID-19 per 1,000 county population.

The independent variables considered for inclusion in the models represent various factors that are hypothesized in this study to play a role in COVID-19 outcomes such as the number of confirmed cases and deaths within a county. These include county-level measures of social distancing and their enforcement severity levels, human mobility, hospital capacity, COVID-19 testing capacity, preventive public health policies as well as county-level socioeconomic and sociodemographic attributes including proportion of the population that is considered to be more vulnerable to COVID-19 (e.g., older adults).

Table 1 lists all the variables that were considered for inclusion in the models along with brief descriptions, descriptive statistics, and data sources.

As it can be seen from Table 1, between January 1, 2020 and June 10, 2020, which is the time period of the study, each day there was on average nearly 1.04 confirmed COVID-19 cases and 0.03 COVID-19 deaths per 1,000 county population in counties comprising the study area (i.e., the DMV area). For the same period, the average daily number of active COVID-19 cases per 1,000 county population was 0.93, whereas the average daily number of COVID-19–exposed residents (i.e., county residents who were already exposed to COVID-19) per 1,000 county population was 3.04. In addition, the table indicates that approximately an average number of 80 external trips per day were made to each county within the DMV area by infected individuals from out of state or county.

With respect to social distancing and human mobility measures, Table 1 shows that for each day within the study time period (January 1 –June 10, 2020) on average nearly 22% of residents of DMV counties stayed at home, and over 19% of the DMV counties' workforce worked from home. Further, the average daily number of person trips by any mode was a little over 3 trips, and the average daily person-miles traveled by all modes was approximately 37 miles. The average transit mode share for the counties within the DMV area was slightly over 2%.

Also, the panel data show that within the study time period (January 1 –June 10, 2020), the average number of days passed since the declaration of the state of emergency and the issuance of the stay-at-home order for the state in which the DMV county located was 26 days and 23 days, respectively. The enforcement severity level for the stay-at-home orders for the DMV counties varied by state. As indicated in Table 1, with a penalty that consisted of confinement in jail for up to one year and a fine of up to $5,000 for violation of the stay-at-home order, Maryland counties were under the strictest enforcement of the stay-at-home order during the study time period (January 1 –June 10, 2020). Compared with Maryland and Virginia where the possible jail time for noncompliance with the stay-at-home order was one year, D.C. had the least strict enforcement level with a possible jail time of only 90 days for violation of the order.

Moreover, the table indicates that within the study time period (January 1 –June 10, 2020), the average number of days passed since the announcement of phase 1 and phase 2 reopening

**Table 1. Model variables: Descriptions and summary statistics.**

| Variable | Description | Mean | Standard Deviation | Computation/Source |
|---|---|---|---|---|
| *COVID-19* | | | | |
| covid-19 confirmed cases* | cumulative number of confirmed covid-19 cases per 1,000 county population | 1.0355 | 2.8301 | calculated based on JHU data [30] |
| covid-19 deaths* | cumulative number of deaths from covid-19 per 1,000 county population | 0.0344 | 0.1156 | calculated based on JHU data [30] |
| active cases | number of active covid-19 cases per 1,000 county population | 0.9305 | 1.3142 | MTI [29] |
| imported cases | number of external trips by infected persons from out of state/county | 79.78645 | 268.6624 | MTI [29] |
| covid-19 exposure | number of county residents already exposed to covid-19 per 1,000 county population | 3.042862 | 4.7249 | MTI [29] |
| days decreasing covid-19 cases | number of days with decreasing covid-19 cases | 13.9264 | 20.4886 | MTI [29] |
| *Social Distancing/Mobility/Preventive Policies* | | | | |
| staying home | percentage of residents staying at home (i.e., no trips with a non-home trip end more than one mile away from home) | 21.9981 | 8.3731 | MTI [29] |
| teleworking | percentage of the county's workforce working from home | 19.3486 | 13.4661 | MTI [29] |
| trips/person | average number of all trips made per person per day | 3.3979 | 0.5343 | MTI [29] |
| miles/person | average person-miles traveled on all modes per person per day | 36.9301 | 12.6596 | MTI [29] |
| transit mode share | percentage of rail and bus transit mode share for the county | 2.2191 | 4.6884 | Census Bureau |
| days since state of emergency | number of days passed since the declaration of the state of emergency for the state | 26.4666 | 30.5836 | web sites for governments of the D.C., MD, and VA |
| days since stay-at-home order | number of days passed since the issuance of the stay-at-home order for the state | 23.2117 | 23.2117 | web sites for governments of the D.C., MD, and VA |
| enforcement severity level | the level of enforcement severity for violating the state's stay-at-home order: | 1.8544 | 0.3701 | web sites for governments of the D.C., MD, and VA |
| | 1 = confinement in jail for ≤ 12 months & a fine of ≤ $5,000, or both (enforcement in Maryland); | | | |
| | 2 = confinement in jail for ≤ 12 months & a fine of ≤ $2,500, or both (enforcement in Virginia); | | | |
| | 3 = confinement in jail for ≤ 90 days & a fine of ≤ $5,000, or both (enforcement in D.C.) | | | |
| days since phase 1 reopening | number of days passed since the announcement of phase 1 reopening for the county | 1.9908 | 5.6203 | web sites for governments of the D.C., MD, and VA |
| days since phase 2 reopening | number of days passed since the announcement of phase 2 reopening for the county | 0.1082 | 0.6764 | web sites for governments of the D.C., MD, and VA |
| days since masks required | number of days passed since the requirement of wearing face coverings/masks in public | 1.8822 | 7.1860 | web sites for governments of the D.C., MD, and VA |
| *Hospital Capacity/Testing* | | | | |
| ventilator shortage | number of ventilators needed for covid-19 patients | 108.9372 | 142.7882 | MTI [29] |
| ICU availability | number of ICU beds per 1,000 county population | 0.2391 | 0.0281 | MTI [29] |
| testing capacity gap | ability to provide enough tests based on WHO-recommended positive test rate proxy (high positive test rates indicate a lack of sufficient testing and testing capacity gap) | 8.7705 | 9.7684 | MTI [29] |
| tests conducted | number of covid-19 tests done per 1,000 county population | 8.6720 | 13.8125 | MTI [29] |
| *Socioeconomic/Sociodemographic/Vulnerable Population* | | | | |
| median income | median household income for the county (in dollars) | 61,143.37 | 21,978.85 | Census Bureau |
| unemployment claim rate | new weekly unemployment insurance claims/1,000 workers | 4.7865 | 4.9218 | Department of Labor |
| change in consumption | percent change in consumption from the pre-pandemic baseline based on observed changes in trips to various types of consumption sites | -3.3823 | 12.2224 | MTI [29] |
| African Americans | percentage of the county population that is African American | 18.7981 | 16.5720 | Census Bureau |
| Hispanic | percentage of the county population that is Hispanic | 5.4335 | 5.5998 | Census Bureau |
| male | percentage of the county population that is male | 49.2864 | 2.4661 | Census Bureau |

*(Continued)*

**Table 1.** (Continued)

| Variable | Description | Mean | Standard Deviation | Computation/Source |
|---|---|---|---|---|
| population over 60 | percentage of the county population over the age of 60 | 25.1075 | 6.3820 | Census Bureau |
| *Other Control Variables* | | | | |
| population density | population density of the county | 869.6835 | 1747.414 | MTI [29] |
| employment density | employment density of the county | 595.1456 | 1628.213 | MTI [29] |
| hot spots | number of points of interests for crowd gathering per 1,000 county population | 130.7278 | 48.3099 | MTI [29] |
| weekend | 1 = day is Saturday or Sunday, 0 = otherwise | — | — | 2020 calendar |

Notes: * indicates dependent variable; JHU = Johns Hopkins University; MTI = Maryland Transportation Institute, University of Maryland.

policies for the DMV counties was 2 days and 0.1 days, respectively. The panel data also show that for the DMV counties within the study time period, the average number of days passed since the requirement of wearing a face mask/covering in public spaces was 1.88 days. The start dates of phase 1 and phase 2 reopening and the date when requirement of wearing a face mask was announced varied by state as well as by county within states depending on local governments' reopening and preventive measure policies.

Regarding hospital capacity and COVID-19 testing measures, Table 1 reveals that during the study time period (January 1 –June 10, 2020), the average daily number of ventilators needed for COVID-19 patients was nearly 109 ventilators, whereas the average daily number of ICU beds per 1,000 county population was 0.24 beds. These statistics may be an indicator of limited access to scarce critical care resources such as ventilators and ICU beds for residents of DMV counties. Also, during the study time period, the average daily number of COVID-19 tests completed per 1,000 county population was nearly 9 tests.

Other descriptive statistics for the data used in this study characterize the study area (i.e., the DMV area) by its socioeconomic and sociodemographic attributes. For example, the table shows that the average annual household median income for DMV counties was slightly over 60,000 dollars, whereas the average number of new weekly unemployment insurance claims per 1,000 county workers was close to 5 claims during the study time period (January 1 –June 10, 2020). Over the same period, the average daily percent change in consumption from the pre-pandemic baseline was -3.4%, meaning that the consumption levels of DMV residents decreased by a daily average of 3.4% after the COVID-19 pandemic. Also, the average percentages of the county population that was African American and Hispanic were approximately 19% and 5%, respectively; the average percentage of the county older adult population (i.e., county population over the age of 60) was 25%; and the average percentage of the county population that was male was approximately 49%. These statistics characterize the population of the DMV counties based on a few risk factors that have been suggested in past research to make certain population more vulnerable to COVID-19 such as being of the African American or Hispanic race, being of older age, and being male [5–9].

Pearson correlation coefficients were calculated to examine correlation between the independent variables. Efforts were made to eliminate highly correlated variables and reduce the risk of multicollinearity in the models. For example, due to a high correlation between variables representing the population density and employment density of the county (r = 0.88), the latter was excluded from the models. Also, the variable representing the number of days passed since the issuance of stay-at-home order was not included in the models because it showed almost no variation across the DMV counties. This was due to D.C., Maryland, and Virginia all issuing their stay-at-home orders at the same time (Maryland and Virginia on March 30, 2020 and D.C. on April 1, 2020).

Further, a two-week lagged variable was included in the models for a few of the variables that potentially contain the effects of the COVID-19 pandemic. These include the COVID-19 active cases and imported cases as well as the staying home, teleworking, and unemployment claim rate variables. The two-week lagged variables reflect the 14-day period for development of COVID-19 symptoms after exposure to the virus. To monitor health status and help prevent spread of the disease, the Centers for Disease Control and Prevention (CDC) recommends a 14-day self-quarantine for individuals who might have been exposed to COVID-19 because symptoms of the disease may take up to 14 days after exposure to the virus to appear [33, 34]. Research has shown that the current 14-day period of active monitoring recommended by the CDC is well supported by empirical evidence as nearly all infected persons are expected to develop symptoms within 12 days of exposure [35].

## Results and discussion

Table 2 summarizes the results of the panel-data models for the daily rate of confirmed COVID-19 cases within the county, and the daily rate of COVID-19 deaths within the county for the DMV area.

The model assumptions are a) presence of first-order autocorrelation within panels (i.e., counties); and b) heteroskedasticity across panels (i.e., counties). The former assumption was considered because the results of the Wooldridge's test for presence of serial correlation in the idiosyncratic error term in panel-data models [32] indicated that the idiosyncratic errors in the models were serially correlated. The serial correlation parameter was assumed to be unique for each panel in the model. This allows each panel to have errors that follow a different first-order autocorrelation process [36]. The generalized least squares (GLS) panel data model, which fits linear panel-data models by using feasible generalized least squares, was employed to address the correlation structure and the assumed heteroskedasticity across panels.

### COVID-19–related measures

As expected, the results show that the 14-day lagged number of active COVID-19 cases per 1,000 county population and the number of county residents already exposed to COVID-19 per 1,000 county population are positively linked with the daily rate of confirmed COVID-19 cases (i.e., infections) and the daily rate of COVID-19 deaths within the county. Further, the 14-day lagged number of external trips to the county by infected persons from out of state/ county also shows a positive link with the daily rate of confirmed COVID-19 cases and the daily rate of COVID-19 deaths within the county. This highlights the importance of travel restrictions and recommendation/requirement of 14-day quarantine period on external trips by some states or local governments in preventing transmission of COVID-19 and related human deaths.

### Social distancing, human mobility, and preventive policy measures

The results of the panel models also provide evidence that measures of social distancing, travel restrictions, and human mobility as well as related preventive measures and policies play a role in COVID-19 outcomes. Interestingly, the 14-day lagged variable on teleworking, which captures the percentage of the county's workforce that worked from home, is negatively related to the daily rates of COVID-19 infections (i.e., confirmed cases) and deaths within the county. This finding is consistent with past research that suggested teleworking can help reduce the risk of exposure to COVID-19 by reducing face-to-face interactions between workers [15].

Moreover, increased daily rates of COVID-19 infections within the county are related to increased numbers of daily person trips by any mode of travel. This result implies that less

**Table 2. Panel model results (generalized least squares model).**

| Dependent Variable / Independent Variable | COVID-19 Confirmed Cases | | COVID-19 Deaths | |
|---|---|---|---|---|
| | Coefficient | p-value | Coefficient | p-value |
| *COVID-19* | | | | |
| active cases (lagged) | 0.008475*** | 0.001 | 0.000242*** | 0.010 |
| imported cases (lagged) | 0.000025*** | 0.000 | 0.000003*** | 0.000 |
| covid-19 exposure | 0.013466*** | 0.000 | 0.000411*** | 0.000 |
| days decreasing covid-19 cases | -0.000843*** | 0.000 | NS | NS |
| *Social Distancing/Mobility/Preventive Policies* | | | | |
| staying home (lagged) | NS | NS | NS | NS |
| teleworking (lagged) | -0.001262*** | 0.000 | -0.000048*** | 0.000 |
| trips/person | 0.012866*** | 0.000 | NS | NS |
| miles/person | NS | NS | NS | NS |
| transit mode share | 0.100582*** | 0.000 | — | — |
| days since state of emergency | 0.018151*** | 0.000 | 0.000788*** | 0.000 |
| enforcement severity level | 0.221426*** | 0.000 | 0.127647*** | 0.000 |
| days since phase 1 reopening | -0.031292*** | 0.000 | -0.000729*** | 0.000 |
| days since phase 2 reopening | -0.020214*** | 0.000 | -0.001331*** | 0.000 |
| days since masks required | -0.032088*** | 0.000 | -0.002357*** | 0.000 |
| *Hospital Capacity/Testing* | | | | |
| ventilator shortage | — | — | 0.000012*** | 0.000 |
| ICU availability | — | — | -7.067309*** | 0.000 |
| testing capacity gap | 0.000874*** | 0.000 | NS | NS |
| tests conducted | 0.040973*** | 0.000 | — | — |
| *Socioeconomic/Sociodemographic/Vulnerable Population* | | | | |
| median income | -0.000002** | 0.048 | NS | NS |
| unemployment claim rate (lagged) | 0.001038** | 0.013 | — | — |
| change in consumption | -0.000339*** | 0.000 | — | — |
| African Americans | 0.004556*** | 0.000 | 0.000390*** | 0.000 |
| Hispanic | 0.026474*** | 0.000 | 0.001019*** | 0.000 |
| male | — | — | NS | NS |
| population over 60 | NS | NS | 0.000564*** | 0.001 |
| *Other Control Variables* | | | | |
| population density | -0.000195*** | 0.000 | — | — |
| hot spots | NS | NS | — | — |
| weekend | 0.005926*** | 0.000 | — | — |
| observations = 25,596; panels (counties) = 158; models' prob. > chi$^2$ = 0.0000 | | | | |

Notes: **, *** = coefficient is significant at the 5% and 1% significance level, respectively; NS = coefficient does not reach the 5% significance level;— = variable not included in the model.

travel by individuals (i.e., less human mobility) can lead to a reduction in transmission of COVID-19—a finding supported by previous studies suggesting that travel restrictions can delay the spread of COVID-19 [26, 27]. Higher levels of traveling by public transportation modes are positively linked with daily rates of COVID-19 infections within the county. This means that usage of public transit can increase the risk of transmission of the disease, which is a reasonable finding considering that using public transit involves being in crowded spaces where practicing social/physical distancing may be difficult.

With respect to preventive policies, a salient finding is the one on the impact of enforcement severity of the stay-at-home orders. The results indicate that as the value of the *enforcement severity level* variable moves towards larger numbers (i.e., the enforcement becomes less severe), the daily rates of COVID-19 infections and deaths within the county increase. This implies that through acting as deterrents, stricter enforcement policies and heavier penalties for noncompliance with stay-at-home orders can help in decreasing the transmission of COVID-19 and related deaths.

Additionally, both the variables representing the number of days passed since the announcement of phased reopening plans for the county (i.e., phase 1 and phase 2) show a negative link with the daily rates of confirmed COVID-19 cases and deaths within the county. This finding can mean that phased reopening plans and policies that consider gradual relaxation of social distancing requirements and travel restrictions can prevent a resurgence of COVID-19 and aid in recovery from it. This is consistent with previous research suggesting that premature relaxation of social distancing measures and travel restrictions can lead to an increase in the number of infections [16]. Therefore, the importance of reopening times in containing the spread of COVID-19 are highlighted through the model results.

Further, the results show that as the number of days since the requirement of wearing face coverings/masks in public increases, the daily rates of COVID-19 infections and COVID-19– related deaths within the county decrease. This is an important finding, which is in line with previous research that highlighted the key role of wearing a face mask in public settings where maintaining social/physical distancing is difficult in slowing the transmission of COVID-19, protecting vulnerable populations such older adults from its worse outcomes, and saving lives [5, 7].

## Hospital capacity and testing capacity measures

Evidence is also provided by model results for the crucial role of hospital capacity and testing capacity measures in COVID-19 outcomes. More specifically, an increased ventilator shortage within the county (i.e., a higher number of ventilators needed for COVID-19 patients) is linked with an increased daily rate of COVID-19 deaths within the county. On the other hand, more availability of ICU beds within the county (i.e., higher numbers of ICU beds per 1,000 county population) is negatively linked with daily rate of COVID-19 deaths within the county. This result is consistent with past empirical evidence showing that COVID-19 mortality rates within the county significantly declined with an increased rate of ICU bed availability for the county population [37]. Overall, these findings are also consistent with past studies suggesting that better access to scarce critical care resources such as ventilators and ICU beds is essential in preventing worse COVID-19 outcomes such as death [20–22].

In addition, the *testing capacity gap* variable is positively related to the daily rate of confirmed COVID-19 cases (i.e., COVID-19 infections) within the county. This result indicates that a lack in sufficient testing and gaps in testing capacity can lead to an increased risk of transmission of COVID-19. Higher daily rates of confirmed COVID-19 cases within the county are also related to higher numbers of COVID-19 tests conducted per 1,000 county population, which is an expected finding; more testing results in detection of more infected cases.

## Socioeconomic/sociodemographic/vulnerable population measures

Table 2 also shows that socioeconomic attributes of the county such as median household income, rate of claims for unemployment insurance, and percent of change in consumption influence COVID-19 outcomes. Higher median household incomes for the county are linked with lower daily rates of confirmed COVID-19 cases. This is an expected result as higher

incomes can be an indicator of the ability and affordability to work remotely, which reduces the number of face-to-face interactions, and thereby can prevent transmission of COVID-19. Previous research suggests that poorer individuals are at higher risk for COVID-19 exposure because they are more likely to live in more crowded households and neighborhoods and are more likely to hold occupations that make it difficult to practice social distancing or teleworking [6, 7]. Other research showed that individuals who were more likely to shift to teleworking following the COVID-19 outbreak were highly educated with high income levels [15]. The findings of the present study provide empirical evidence that income levels can play a significant role in spread of COVID-19.

Other county-level socioeconomic factors also affect the spread of COVID-19. The rate of claims for unemployment insurance within the county (i.e., the number of new weekly unemployment insurance claims per 1,000 county workers) is positively linked with the daily rate of confirmed COVID-19 cases within the county. This finding implies that having more unemployed residents can lead to an increase in the spread of COVID-19 within the county. One reason for this can be that newly-unemployed individuals potentially gain some spare time, which they may spend on running long-overdue errands or visiting friends/family. This can increase their risk of contracting the disease and transmitting it, if these activities are not essential or if they are not performed in accordance with CDC's recommendations and guidelines on social/physical distancing and running essential errands [12, 38]. Conversely, a higher percent change in consumption from the pre-pandemic baseline for the county has a negative effect on the daily rate of confirmed COVID-19 cases within the county. This means that lower levels of good consumption can reduce the spread of COVID-19. As this variable is computed based on observed changes in trips to various types of consumption sites [29], this finding reemphasizes the crucial role of reduced levels of travel and human mobility in slowing the transmission of COVID-19.

Based on the model estimates, other influential factors in COVID-19 outcomes for the county are sociodemographic attributes of the county including the percentages of the county population that are: African American, Hispanic, and older adults (i.e., over the age of 60 years). These characteristics have been suggested by several studies to be vulnerability factors for more severe COVID-19 outcomes [5–8].

The results show that higher percentages of African American or Hispanic population within the county are linked with higher daily rates of confirmed COVID-19 cases and COVID-19–related deaths within the county. These results are in line with COVID-19 statistics for states within the DMV area. Table 3 shows the percentage of COVID-19 confirmed cases and deaths involving African Americans and Hispanic/Latinos as well as the percentage of the state's population that these minorities constitute for the DMV area.

As seen from Table 3 approximately 48% of confirmed COVID-19 cases and 75% of COVID-19 deaths in D.C., 34% of confirmed COVID-19 cases and 37% of COVID-19 deaths

**Table 3. Percentage of COVID-19 cases/deaths and population for African Americans and Hispanic/Latinos by DMV state.**

|  | African Americans | | | Hispanic or Latinos | | |
|---|---|---|---|---|---|---|
|  | % COVID-19 Cases | % COVID-19 Deaths | % of Population | % COVID-19 Cases | % COVID-19 Deaths | % of Population |
| **District of Columbia** | 48% | 75% | 46% | 24% | 13% | 11% |
| **Maryland** | 34% | 37% | 29% | 20% | 10% | 10% |
| **Virginia** | 21% | 24% | 19% | 20% | 8% | 9% |

Source of data: COVID Tracking Project Racial Data Tracker (https://covidtracking.com/race/dashboard).

Data as of January 10, 2020.

in Maryland, and 21% of confirmed COVID-19 cases and 24% of COVID-19 deaths in Virginia involved African Americans, whereas African Americans constitute only about 46%, 29%, and 19% of the population of these states, respectively. Also, approximately 24% of confirmed COVID-19 cases and 13% of COVID-19 deaths in D.C., 20% of confirmed COVID-19 cases and 10% of COVID-19 deaths in Maryland, and 20% of confirmed COVID-19 cases and 8% of COVID-19 deaths in Virginia involved Hispanic or Latinos, whereas Hispanic or Latinos made up only about 11%, 10%, and 9% of the population of these states, respectively. Generally speaking, these statistics indicate that in the DMV states, the African American and Hispanic/Latino population account for higher percentages of confirmed COVID-19 cases and related deaths than their representation percentages in the population.

The findings of the present study with respect to the effect of race on COVID-19 outcomes are also supported by previous research suggesting that certain population groups including African Americans and Hispanics have been disproportionately affected by the COVID-19 pandemic [5–7, 9, 39, 40]. Many reasons can contribute to such findings. Among these can be factors related to a lower socioeconomic status such as lack of or limited access to health insurance and healthcare, lack of or limited access to healthy food, living in multigenerational and crowded houses, and holding occupations in which maintaining social distancing and practicing teleworking is difficult or impractical (e.g., retail and grocery stores, services). A previous study found that white, highly educated, and high-income individuals were more likely to switch to teleworking and to maintain employment following the COVID-19 outbreak in the U.S. [15]. Other research also suggests that low socioeconomic status puts underrepresented minorities at greater risk for COVID-19 exposure and mortality [6, 7, 40], partly due to inability to practice teleworking or to maintain social distancing, which is one of the most effective strategies known to reduce risk of COVID-19 infection [7].

Furthermore, a higher daily rate of COVID-19 deaths within the county is positively related to a higher percentage of the county population that is over the age of 60 years. This implies that older adults can be at increased risk of worse COVID-19–related outcomes including death. This result corroborates empirical evidence from a previous study that found counties with a higher percentage of the population over age 60 had higher COVID-19 mortality rates [37]. This finding is also in line with other past research suggesting that older adults have a higher risk of severe COVID-19 outcomes such as elevated rates of hospitalization and death [5–8, 39]. With regards to gender, some studies suggest that men are at higher risk for severe COVID-19 outcomes such as death [10, 39]. However, the coefficient estimate of the *male* variable in the present study, which represents the percentage of the county population that is male, does not reach the 5% significance level in the model for the daily rate of COVID-19 deaths within the county.

The results also indicate that population density is negatively linked with daily rates of COVID-19 infections. This result may seem counter-intuitive because in theory, denser areas are expected to increase the transmission of the disease due to facilitation of human interactions. One reason for this finding, however, can be that denser areas are typically urban settings that provide more and higher levels of services such as home delivery services, and can thereby facilitate the practice of social distancing [37]. While it has been suggested in previous research that higher densities may act as a risk factor for COVID-19 [7], empirical evidence on the role of density measures such as population, employment, or activity densities in COVID-19 outcomes is still scarce. One study that investigated the effect of density on county-level COVID-19 outcomes reported the effect of density on COVID-19 infection rate within the county statistically insignificant [37]. Therefore, additional research may be needed to probe the link between density measures and COVID-19 outcomes.

## Conclusions

Understanding the factors that play a role in the outcomes of a highly contagious disease and accurately predicting those outcomes is critical for the containment of COVID-19 and any other future contagion. The COVID-19 pandemic has been a rapidly evolving situation and many gaps exist in empirical research on the factors that affect the spread and outcomes of this new and ongoing public health threat. To help fill the gap in empirical knowledge on COVID-19 outcomes, this study uses longitudinal data—including big location-based service data providing information on social distancing and mobility trends—to model the daily rates of confirmed COVID-19 cases and COVID-19–related deaths in District of Columbia as well as counties located in the states of Maryland and Virginia. Through these models, the effects of several factors that have—in theory—been suggested to influence the COVID-19 outcomes are empirically tested. These factors include measures of social distancing, human mobility, preventive policies, enforcement severity, and hospital as well as testing capacities.

The results provide evidence that social distancing and reduced human mobility contribute to slowing the spread of COVID-19 and preventing its worse outcomes such as death. Lower rates of COVID-19 infections and/or deaths are found to be linked with higher levels of social distancing and lower levels of travel through measures such increased teleworking, fewer external (i.e., out-of-county/state) trips by infected individuals, and fewer trips by all modes of travel—especially, the public transit mode. These findings enhance the arguments that promote implementation of social distancing preventive measures and travel restrictions to effectively mitigate the effects of a global pandemic such as COVID-19.

Other related preventive measures and policies including enforcement strategies of stay-at-home orders, time of phased reopening plans, and requirement of wearing a face mask in public settings are also found to be key factors in COVID-19 outcomes. Particularly, lower rates of COVID-19 infections and deaths are linked with stricter enforcement and more severe penalties for noncompliance with stay-at-home orders. This finding highlights the importance of enforcement of government-imposed restrictions as a preventive tool in mitigating the impacts of the COVID-19 pandemic in the U.S. The results of the study further suggest that phased reopening plans that focus on gradual relaxation of social distancing measures and travel restrictions can prevent a resurgence of COVID-19 as lower rates of COVID-19 infections and deaths are related to phase 1 and phase 2 reopening times for the counties within the study area. Together, these findings can assist in designing enforcement strategies and reopening plans that can deepen the benefits of social distancing measures.

In addition, the critical role of wearing a face mask in containing the spread of COVID-19 and preventing human deaths are reemphasized through the findings of this study. Lower rates of COVID-19 infections and deaths within the county are found to be linked with the number of days since wearing a face mask/covering became required for the county residents. Although this finding is not surprising as using a face mask is considered to be an effective strategy in preventing the contraction of a communicable respiratory disease such as COVID-19 [41], the empirical evidence lends an additional layer of confidence to the argument.

Moreover, higher rates of COVID-19 infections are found to be related to a lack of sufficient testing and testing capacity gaps. This finding underscores the important role of adequate testing during a mounting pandemic in slowing the spread of the disease. Additionally, the study finds that increased hospital capacity through increased availability of ventilators and ICU beds can prevent COVID-19–related deaths. These findings can assist in planning and prioritization of scarce healthcare resources during a pandemic and avoiding the dire consequences of a lack of access to such critical equipment.

The findings also indicate that higher COVID-19 infection rates are linked with lower income levels and higher unemployment rates. Further, the study provides additional empirical evidence for the reports that certain minority groups such as African Americans and Hispanics bear a disproportionate burden of COVID-19–related outcomes. These health-related disparities can be attributable to disparities in socioeconomic status. Lack of or limited access to healthcare, healthy food, digital devices and the Internet, as well as overcrowded living conditions and employment in sectors that make social distancing more challenging to practice are among the disadvantages faced by these minority populations that may leave them vulnerable to a global pandemic such as COVID-19. These findings indicate that while the COVID-19 pandemic is affecting everyone in the U.S., it is not affecting everyone equally. Such findings provide an opportunity for decision-makers to address the health disparities among minority groups by implementing public policies that can narrow the gap between health and wealth of different racial groups in the U.S. Such policies can help protect all U.S. population against new waves of COVID-19, its mutated variants, or any potential future pandemic whose nature is similar to that of COVID-19. The findings also corroborate previous research indicating that older adults are among other vulnerable groups who are at higher risk of worse COVID-19 outcomes including death.

Population density is found to be negatively linked with COVID-19 infection rates. This finding can be an indication of facilitated practice of social distancing that denser areas can offer through providing better levels of services such as home deliveries [37]. Nonetheless, this finding warrants further research since there is limited empirical evidence on the role of density in COVID-19 outcomes.

Together, the study findings provide a deeper understanding of the role of factors such as social distancing, mobility, preventive policies, and hospital as well as testing capacity in COVID-19 outcomes in megaregions such as the DMV area. Particularly, increased social distancing, reduced mobility, stricter enforcement, careful planning of reopening times, mandating the usage of a face mask/covering, and increased hospital and testing capacities can slow the spread of COVID-19 and save lives. These empirical findings can be generalized to other U.S. megaregions with similar characteristics of the DMV area.

This study has a few limitations. First, the relationship between urban form and COVID-19 outcomes has not been thoroughly examined in this study. Although the analysis includes a population density variable and a variable capturing the rate of hot spots within the county, many other measures of urban form with a potential to impact COVID-19 outcomes have not been included. These include measures of access to healthcare, access to healthy food, and access to clean air. Future work can probe the effects of such factors in COVID-19 outcomes.

Second, underlying medical conditions such as hypertension, diabetes, chronic lung disease, asthma, cardiovascular disease, obesity, and cancer increase the risk of severe illness from COVID-19 [25]. The effect of measures representing such factors on COVID-19 outcomes within an area can be examined in future research. Among other possible underlying causes of increased susceptibility to COVID-19 or its worse outcomes are genetic predisposition. Future work can investigate the role of genetic factors in contraction and severity of COVID-19 at an individual level.

Further, future research can benefit from inclusion of various other factors with a potential to influence COVID-19 outcomes within megaregions in the analysis. Among these can be measures characterizing teleshopping trends and contact tracing practices. Big data such as the location-based service data used in this study offer a promising potential to be utilized in investigation of the role of teleshopping changes during the pandemic in COVID-19 outcomes within megaregions.

Lastly, the rates of confirmed COVID-19 cases and deaths have been modeled separately in this study, assuming there is no endogeneity in the analysis. Future work can benefit from examining endogeneity issues in modeling COVID-19 outcomes such as rates of infections and deaths.

The COVID-19 pandemic continues to rampage across the world. The availability of several types of vaccines is a crucial new development, which promises protection against the disease through herd immunity in the future. Nonetheless, absent effective vaccination plans, and until herd immunity is reached through vaccination of the entire population, the only method to protect against the 2019-nCoV coronavirus and its mutant strains is to prevent transmission and contraction of it. The main contribution of this study is adding to the body of empirical knowledge on the link between various preventive measures including social distancing, restricted human mobility, and policy measures that are essential to slow the spread of this disease and prevent deaths cause by it.

The study findings can assist decision-makers in designing and implementing the most effective policies and preventive response strategies against COVID-19, and providing clear and concise directions to the public to protect themselves and others. Such efforts can minimize the loss of human life, optimize the recovery plans, and expedite the return to normalcy during the current pandemic and any potential future one.

## Acknowledgments

We would like to thank and acknowledge our partners and data sources in this effort: (1) various mobile device location data providers; (2) Amazon Web Service and its Senior Solutions Architect, Jianjun Xu, for providing cloud computing and technical support; (3) computational algorithms developed and validated in a previous USDOT Federal Highway Administration's Exploratory Advanced Research Program project; and (4) COVID-19 confirmed case data from the Johns Hopkins University Github repository and sociodemographic data from the U.S. Census Bureau.

## Author Contributions

**Conceptualization:** Jina Mahmoudi, Chenfeng Xiong.

**Data curation:** Jina Mahmoudi, Chenfeng Xiong.

**Formal analysis:** Jina Mahmoudi, Chenfeng Xiong.

**Methodology:** Jina Mahmoudi, Chenfeng Xiong.

**Supervision:** Chenfeng Xiong.

**Writing – original draft:** Jina Mahmoudi, Chenfeng Xiong.

**Writing – review & editing:** Jina Mahmoudi, Chenfeng Xiong.

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
