## [Decision Letter · Decision Letter 0]

24 Nov 2021

PONE-D-21-15442

How social distancing, mobility, and preventive policies affect COVID-19 outcomes: Big data-driven evidence from the District of Columbia-Maryland-Virginia (DMV) megaregion

PLOS ONE

Dear Dr. Xiong,

Thank you for submitting your manuscript to PLOS ONE. After careful consideration, we feel that it has merit but does not fully meet PLOS ONE’s publication criteria as it currently stands. Therefore, we invite you to submit a revised version of the manuscript that addresses the points raised during the review process.

We look forward to receiving your revised manuscript.

Kind regards,

Zezhi Li, Ph.D., M.D.

Academic Editor

PLOS ONE

Journal Requirements:

3. PLOS requires an ORCID iD for the corresponding author in Editorial Manager on papers submitted after December 6th, 2016. Please ensure that you have an ORCID iD and that it is validated in Editorial Manager. To do this, go to ‘Update my Information’ (in the upper left-hand corner of the main menu), and click on the Fetch/Validate link next to the ORCID field. This will take you to the ORCID site and allow you to create a new iD or authenticate a pre-existing iD in Editorial Manager. Please see the following video for instructions on linking an ORCID iD to your Editorial Manager account: https://www.youtube.com/watch?v=_xcclfuvtxQ.

4. We note that Figure 3 & 4  in your submission contain [map/satellite] images which may be copyrighted. All PLOS content is published under the Creative Commons Attribution License (CC BY 4.0), which means that the manuscript, images, and Supporting Information files will be freely available online, and any third party is permitted to access, download, copy, distribute, and use these materials in any way, even commercially, with proper attribution. For these reasons, we cannot publish previously copyrighted maps or satellite images created using proprietary data, such as Google software (Google Maps, Street View, and Earth). For more information, see our copyright guidelines: http://journals.plos.org/plosone/s/licenses-and-copyright.

 a. You may seek permission from the original copyright holder of Figures 3 & 4 to publish the content specifically under the CC BY 4.0 license. 

Reviewers' comments:

Reviewer's Responses to Questions

**Comments to the Author**

1. Is the manuscript technically sound, and do the data support the conclusions?

Reviewer #1: Yes

2. Has the statistical analysis been performed appropriately and rigorously? 

Reviewer #1: Yes

3. Have the authors made all data underlying the findings in their manuscript fully available?

Reviewer #1: Yes

4. Is the manuscript presented in an intelligible fashion and written in standard English?

Reviewer #1: Yes

5. Review Comments to the Author

Reviewer #1: COVID-19 has infected more than 188 million people and results in more than 4 million deaths since its first report. The pandemic leads to enormous damage to human health and the global economy. Social distancing, wearing face coverings and restricting travels have proven to be effective in slowing the spread of infections. The effects of human mobility restrictions on the spread of COVID-19 have also been widely investigated around the world. This study provides a link between human mobility changes and COVID-19 infections within a megaregion that contains the U.S. capital city. Several concerns need to be addressed.

1. How many mobile devices were included in this study and how to exclude the possible duplicated calculations as they are anonymized devices.

2. is there a specific standard for the “severe penalties” for violating stay-at-home orders which are linked to lower COVID-19 infection?

3. Both the mobilities within communities and long-distance travel play an important role

in the spreading of the pandemic. Did the authors distinguish the two kinds of mobilities in this study?

4. Line 216. The wording is not accurate. there have been a lot of studies revealed the effects of human mobility on the virus spread based on large scale dataset, for example, Ying Zhou et al, 2020, Lancet Digital Health; Moritz U. G. Kraemer et al, 2020, Science; Pierre Nouvellet et al, 2021, Nature Communications.

6. PLOS authors have the option to publish the peer review history of their article (what does this mean?). If published, this will include your full peer review and any attached files.

Reviewer #1: No

---

## [Author Response · Author response to Decision Letter 0]

2 Jan 2022

Please see the "response to reviewers.docx" for a detailed reply to each reviewer/editor comment.

---

## [Decision Letter · Decision Letter 1]

28 Jan 2022

How social distancing, mobility, and preventive policies affect COVID-19 outcomes: Big data-driven evidence from the District of Columbia-Maryland-Virginia (DMV) megaregion

PONE-D-21-15442R1

Dear Dr. Xiong,

We’re pleased to inform you that your manuscript has been judged scientifically suitable for publication and will be formally accepted for publication once it meets all outstanding technical requirements.

Kind regards,

Zezhi Li, Ph.D., M.D.

Academic Editor

PLOS ONE

Additional Editor Comments (optional):

Reviewers' comments:

Reviewer's Responses to Questions

**Comments to the Author**

1. If the authors have adequately addressed your comments raised in a previous round of review and you feel that this manuscript is now acceptable for publication, you may indicate that here to bypass the “Comments to the Author” section, enter your conflict of interest statement in the “Confidential to Editor” section, and submit your "Accept" recommendation.

Reviewer #1: All comments have been addressed

2. Is the manuscript technically sound, and do the data support the conclusions?

Reviewer #1: Yes

3. Has the statistical analysis been performed appropriately and rigorously? 

Reviewer #1: Yes

4. Have the authors made all data underlying the findings in their manuscript fully available?

Reviewer #1: Yes

5. Is the manuscript presented in an intelligible fashion and written in standard English?

Reviewer #1: Yes

6. Review Comments to the Author

Reviewer #1: (No Response)

7. PLOS authors have the option to publish the peer review history of their article (what does this mean?). If published, this will include your full peer review and any attached files.

Reviewer #1: No

---

## [Editor Report · Acceptance letter]

8 Feb 2022

PONE-D-21-15442R1 

How social distancing, mobility, and preventive policies affect COVID-19 outcomes: Big data-driven evidence from the District of Columbia-Maryland-Virginia (DMV) megaregion 

Dear Dr. Xiong:

I'm pleased to inform you that your manuscript has been deemed suitable for publication in PLOS ONE. Congratulations! Your manuscript is now with our production department. 

Kind regards, 

on behalf of

Dr. Zezhi Li 

Academic Editor

PLOS ONE